# Sharp Attention for Sequence to Sequence Learning

## Abstract

Attention mechanism has been widely applied to tasks that output some sequence from an input image. Its success comes from the ability to align relevant parts of the encoded image with the target output. However, most of the existing methods fail to build clear alignment because the aligned parts are unable to well represent the target. In this paper we seek clear alignment in attention mechanism through a *sharpener* module. Since it deliberately locates the target in an image region and refines representation to be target-specific, the alignment and interpretability of attention can be significantly improved. Experiments on synthetic handwritten digit as well as real-world scene text recognition datasets show that our approach outperforms the mainstream ones such as soft and hard attention.

## 1 Introduction

In modern sequence to sequence learning, attention mechanism has become a key building block, because it helps identify relevant parts of the input sequence and align them with the target output at each time step. Such alignment resembles fixation in human vision, where only the object of interest falls on the fovea in the retina, leading the visual scene outside to be largely ignored. Thanks to such ability to select perceptual information, attention mechanism has been successfully applied to many visual tasks, such as scene text recognition (Shi et al., 2019) and image captioning (Xu et al., 2015).

Although a variety of attention mechanisms have been proposed to build alignment, most of them fail to achieve clear alignment. Soft attention (Bahdanau et al., 2015; Luong et al., 2015), the most popular one, aligns a weighted average of the input sequence with the target output throughout the time. Since the weights are never zeros, irrelevant parts are inevitably involved in the alignment and may introduce distraction. For distinct alignment, hard attention (Xu et al., 2015) enforces exactly one input part is employed, regardless of whether it represents the target or not, and thus may still suffer from irrelevant parts. Besides, existing attention mechanisms often regard the input sequence as fixed during alignment establishment. If the target representation given by the selected part(s) is poor, there is no way to fix it. This is especially the case in visual sequence learning, where features are precomputed by a convolutional neural network (CNN) before being fed into attention mechanisms. As each feature only characterises a local fixed image region (i.e. the receptive field), it hardly covers the appearance of the target exactly, thus leading to noisy representation. See Fig. 1(b) for an example.

In this work we address the construction of clear alignment in attention mechanism. This is achieved by aligning the target output with image regions instead of features, which is a more natural approach to alignment for visual sequence learning. A *sharpener* module is then used to make the aligned region as specific as possible to the target, essentially a clear alignment. While it can take any form, the module explored in this paper consists of a localiser and an encoder. The former locates the target in the region, while the latter extracts features from the result for alignment. It is such accurate and specific representation that makes attention mechanism able to pay close attention to the target, leading to improved alignment and thus interpretability. The sharpener can be trained along with any sequence-to-sequence (Seq2Seq) model through back-propagation without extra supervision. Nonetheless, it is also possible to guide its training to further improve alignment quality if auxiliary information is available (see Sect. 4.1). Therefore, the sharpener naturally lends itself to direct attention manipulation, which is yet not available in most of the existing attention mechanisms.

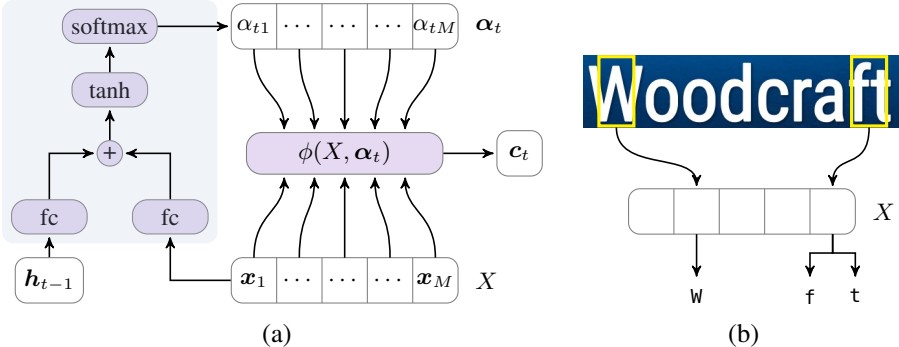

(a)                                    (b)

Figure 1: (a) An outline of the attention mechanism. A query vector $\boldsymbol{h}_{t-1}$ is compared with an input sequence $X$ to figure out where to look via a multilayer perceptron (see the grey box), leading to a set of weights $\boldsymbol{\alpha}_t$. A context vector $\boldsymbol{c}_t$ is then computed for alignment by a function $\phi$, where attention mechanisms generally differ. Soft attention computes $\boldsymbol{c}_t$ as a weighted average of $X$ while hard attention does it by randomly sampling an element from $X$. (b) Poor target representation in attention mechanism. Each element of $X$ only represents a fixed region defined by the receptive field (see yellow boxes). It is hard to accurately represent the target without the distraction of redundant or missing parts (see letters 'f' and 'W' for examples).

## 2 SHARP ATTENTION

### 2.1 ALIGNMENT

Let $X = \{\boldsymbol{x}_1, \ldots, \boldsymbol{x}_M\}$ be an input sequence of length $M$, where $\boldsymbol{x}_i \in \mathbb{R}^D$ is a feature vector representing a region of an input image. Usually, $X$ comes from the last convolutional layer of a backbone network, and $\boldsymbol{x}_i$ delineates a region of fixed size given by the receptive field of that layer. Similarly, we denote the output sequence of length $N$ as $Y = \{\boldsymbol{y}_1, \ldots, \boldsymbol{y}_N\}$, where $\boldsymbol{y}_t \in \mathbb{R}^K$ is a one-of-$K$ encoded vector indicating a discrete token in a vocabulary of size $K$. Our goal is to learn a model that can accurately predict $\boldsymbol{y}_t$ by choosing appropriate $\boldsymbol{x}_i$ in a sequential process. This implicitly asks for building an alignment between $X$ and $Y$ at each time step. To facilitate the construction, we introduce a latent variable $A = \{\boldsymbol{a}_1, \ldots, \boldsymbol{a}_N\}$, where $\boldsymbol{a}_t \in \mathbb{R}^M$ is a one-of-$M$ encoded vector indicating the index of the selected feature vector at some time. For example, $a_{ti} = 1$ refers to the $i$-th element of $X$ (i.e. $\boldsymbol{x}_i$) being chosen to predict $\boldsymbol{y}_t$ at time $t$. Usually, the selected feature is called context vector and denoted as $\boldsymbol{c}_t$.

To learn the model, we maximise a conditional probability

$$p(Y|X) = \sum_A p(Y, A|X) = \sum_A \prod_{t=1}^N p(\boldsymbol{y}_t, A|\underbrace{\boldsymbol{y}_1, \ldots, \boldsymbol{y}_{t-1}}_{\boldsymbol{y}_{<t}}, X) = \prod_{t=1}^N \sum_{\boldsymbol{a}_t} p(\boldsymbol{y}_t, \boldsymbol{a}_t|\boldsymbol{y}_{<t}, X) \quad (1)$$

$$= \prod_{t=1}^N \sum_{\boldsymbol{a}_t} p(\boldsymbol{a}_t|\boldsymbol{y}_{<t}, X) p(\boldsymbol{y}_t|\boldsymbol{y}_{<t}, X, \boldsymbol{a}_t) \equiv \prod_{t=1}^N \sum_{i=1}^M p(a_{ti} = 1|\boldsymbol{y}_{<t}, X) p(\boldsymbol{y}_t|\boldsymbol{y}_{<t}, \boldsymbol{x}_i) \, (2)$$

where the last two terms in (1) are obtained by applying chain rule on $Y$, and by using the assumption that $\boldsymbol{y}_t$ only depends on $\boldsymbol{a}_t$ at time $t$, respectively. Equation (2) clearly defines two major components to compute $p(Y|X)$. One is the chance of selecting each element of $X$, and the other is the likelihood of the target token given the selection. However, the computation of the latter is impractical when $M$ is large because every element of $X$ has to be considered. Two typical approximations to (2) are thus proposed, leading to the soft and hard attention mechanisms.

By using the first order Taylor expansion,[1] we obtain the loss function for soft attention,

$$\log p(Y|X) = \sum_{t=1}^N \log \left( \sum_{i=1}^M \alpha_{ti} p(\boldsymbol{y}_t|\boldsymbol{y}_{<t}, \boldsymbol{x}_i) \right) \approx \sum_{t=1}^N \log p \left( \boldsymbol{y}_t|\boldsymbol{y}_{<t}, \sum_{i=1}^M \alpha_{ti} \boldsymbol{x}_i \right), \quad (3)$$

---

[1] Let $f(\cdot)$ be a function of some random variable. By using Taylor's theorem, the first-order approximation to the expectation $\mathbb{E}[f(\cdot)]$ is given by $\mathbb{E}[f(\cdot)] \approx f(\mathbb{E}[\cdot])$.

64 where we define $\alpha_{ti} \equiv p(a_{ti} = 1|\boldsymbol{y}_{<t}, X)$ to simplify the notation. In (3), the context vector $\boldsymbol{c}_t$ is
65 given by $\sum_{i=1}^{M} \alpha_{ti}\boldsymbol{x}_i$, which means that $\boldsymbol{y}_t$ is no longer predicted by a single element of $X$ but rather
66 a weighted average of $X$, thus leading to the break in alignment. To pursue the alignment such that
67 each target token only depends on one element of $X$, hard attention instead estimates a variational
68 lower bound on $\log p(Y|X)$ using Jensen's inequality,

$$\log p(Y|X) = \sum_{t=1}^{N} \log \left( \sum_{i=1}^{M} \alpha_{ti} p(\boldsymbol{y}_t|\boldsymbol{y}_{<t}, \boldsymbol{x}_i) \right) \geqslant \sum_{t=1}^{N} \sum_{i=1}^{M} \alpha_{ti} \log p(\boldsymbol{y}_t|\boldsymbol{y}_{<t}, \boldsymbol{x}_i). \tag{4}$$

69 Now the optimisation of $\log p(Y|X)$ can be thought as repeating the following steps until happy:
70 (i) estimating $p(\boldsymbol{a}_t|\boldsymbol{y}_{<t}, X)$ by fixing all model parameters; (ii) modifying the parameters to maximise
71 $\log p(\boldsymbol{y}_t|\boldsymbol{y}_{<t}, \boldsymbol{x}_i)$ using each $\boldsymbol{x}_i$. The underlying idea is to increase $\log p(Y|X)$ by iteratively raising
72 the lower bound. Since it is infeasible to consider every $\boldsymbol{x}_i$ as aforementioned, approximation is often
73 adopted and achieved by Monte Carlo sampling (See Sect. 2.4 for details), thus leading $\boldsymbol{c}_t$ to be the
74 sampled feature vector for hard attention.

## 2.2 LOSS FUNCTION

76 Intuitively, if each $\boldsymbol{x}_i$ is an accurate representation of the target token when optimising (4), particularly
77 in the second step, there would be a tight gap between $\log p(Y|X)$ and its lower bound. In contrast,
78 if any poor $\boldsymbol{x}_i$ occurs, the gap may become large and thus result in performance degradation. Subject
79 to the fixed local region, it is unlikely for $\boldsymbol{x}_i$ to well characterise the target token, which makes hard
80 attention hardly achieve clear alignment (see Fig. 1(b)). This motivates us to reformulate (4) for more
81 flexible representation of the target tokens.

82 Suppose we can break down the input image into a set of $M$ local regions, each of which is sufficiently
83 large to cover objects of interest. We would like to maximise the marginal log-likelihood $\log p(Y|R)$,
84 where $R = \{r_1, \ldots, r_M\}$ is the set of regions. Similarly, its lower bound $\ell$ is given by

$$\log p(Y|R) \geqslant \sum_{t=1}^{N} \sum_{\boldsymbol{a}_t} p(\boldsymbol{a}_t|\boldsymbol{y}_{<t}, R) \log p(\boldsymbol{y}_t|\boldsymbol{y}_{<t}, R, \boldsymbol{a}_t) \equiv \ell, \tag{5}$$

85 where $\boldsymbol{a}_t$ is still a one-of-$M$ encoded vector but now refers to the index of the selected region at time
86 $t$. By working with (5), we are not restricted to the representation given by $X$ any more. Consider
87 $\boldsymbol{x}$ to be a function of $r$ parameterised by the backbone network, e.g., $\boldsymbol{x} = f_g(r; \boldsymbol{\theta}_g)$, where $\boldsymbol{\theta}_g$
88 denotes all the weights in the network. By plugging $X = \{f_g(r_1; \boldsymbol{\theta}_g), \ldots, f_g(r_M; \boldsymbol{\theta}_g)\}$ into (4), it
89 is easy to see that the lower bound of $\log p(Y|X)$ is equivalent to that of $\log p(Y|R)$ when partially
90 parameterising the two terms $p(\boldsymbol{a}_t|\boldsymbol{y}_{<t}, R)$ and $\log p(\boldsymbol{y}_t|\boldsymbol{y}_{<t}, R, \boldsymbol{a}_t)$ using $\boldsymbol{\theta}_g$. As $\ell$ is valid for all
91 model parameters, it does not rely on any specific modelling. This allows us to separately model the
92 use of $R$ in each term. For example, we may leverage the same backbone network for the first term to
93 get a rough idea on where to look, while deliberately design a network for the second term to sharpen
94 the focus. It is such flexible parameterisation that makes the construction of clear alignment possible.
95 Below we elaborate the modelling of each term in (5).

## 2.3 MODELLING

97 We use a variant of VGG (Shi et al., 2017) as the backbone network to not only create the set of
98 regions but also process it in $p(\boldsymbol{a}_t|\boldsymbol{y}_{<t}, R)$. When the backbone network is a CNN, we may take
99 advantage of the implicitly defined sliding widow for region generation. For example, our backbone
100 network effectively divides an input image of size $100{\times}32$ into 24 $86{\times}46$ regions when extracting
101 features from the final layer (Araujo et al., 2019). Note that the generation of $R$ is arbitrary and we
102 just use the sliding window for simplicity. To emphasise clear alignment with $\log p(\boldsymbol{y}_t|\boldsymbol{y}_{<t}, R, \boldsymbol{a}_t)$,
103 we leverage a sharpener module that consists of a localiser and an encoder. The former seeks the
104 target in the selected region while the latter extracts features from the result. While a natural choice
105 of the localiser is object detectors, we instead resort to spatial transformer networks (STNs, Jaderberg
106 et al. (2015)) for both computational and labelling efficiency. STN is a lightweight CNN that is able
107 to crop and transform an image region. Its training does not need expensive annotations such as
108 bounding boxes, which are usually unavailable in sequential learning tasks. The encoder can take
109 any form and we use the same CNN to the backbone to simplify the implementation. Let $Z$ be the

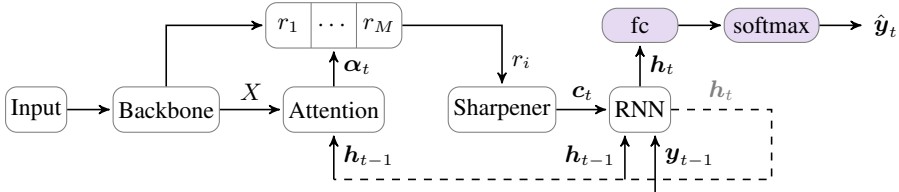

Figure 2: A generic Seq2Seq learning architecture with sharp attention. Given an input image, a backbone network breaks down it into a set of regions and extracts features from each region, leading to a sequence of feature vectors $X$. An attention mechanism (see Fig. 1(a)) uses $X$ and a hidden state vector $\boldsymbol{h}_{t-1}$ to compute a categorical distribution $\boldsymbol{\alpha}_t$, based on which a region is randomly chosen and fed into a sharpener module to compute the context vector $\boldsymbol{c}_t$ for clear alignment. A recurrent neural network (RNN) takes in $\boldsymbol{h}_{t-1}$, $\boldsymbol{c}_t$ and $\boldsymbol{y}_{t-1}$ (the token at previous time step) to update its internal state, and then outputs current $\boldsymbol{h}_t$ for token prediction and next iteration (dashed line).

encoder output, which is also a sequence of feature vectors similar to $X$ with length dependent on the size of the localisation result. The output of the sharpener is the context vector, whose computation will be detailed in Sect. 2.5.

The dependency of the current token $\boldsymbol{y}_t$ on all previous ones $\boldsymbol{y}_{<t}$ over the time is often modelled by an RNN, e.g., long short-term memory (LSTM, Hochreiter & Schmidhuber (1997)). Specifically, it is defined by[2]

$$\boldsymbol{h}_t = f_r(\boldsymbol{h}_{t-1}, \boldsymbol{y}_{t-1}, \boldsymbol{c}_t; \boldsymbol{\theta}_r), \quad \boldsymbol{h}_0 \equiv f_i\left(\frac{1}{M}\sum_{i=1}^{M}\boldsymbol{x}_i; \boldsymbol{\theta}_i\right), \quad \boldsymbol{y}_0 \equiv \boldsymbol{0}, \tag{6}$$

where $f_r(\cdot)$ refers to the non-linear function defined by LSTM with parameters $\boldsymbol{\theta}_r$, $\boldsymbol{h}_t$ is a hidden state vector at time $t$ that summarises the history tokens $\boldsymbol{y}_{<t}$ and is initialised by a fully connected layer $f_i(\cdot)$ that takes an average of $X$ as the input, and a zero vector is used as the initial token $\boldsymbol{y}_0$.

Now we are ready to define the two terms in (5). As we use the backbone network to process $R$ in $p(\boldsymbol{a}_t|\boldsymbol{y}_{<t}, R)$, the probability of selecting a particular region is given by

$$\alpha_{ti} \equiv p(a_{ti} = 1|\boldsymbol{y}_{<t}, R) \equiv p(a_{ti} = 1|\boldsymbol{h}_{t-1}, X) = \text{softmax}(f_a(\boldsymbol{x}_i, \boldsymbol{h}_{t-1}; \boldsymbol{\theta}_a)), \tag{7}$$

where $f_a(\cdot)$ is the attention function as described in Bahdanau et al. (2015) (Fig. 1(a)). The probability of the output token $\hat{\boldsymbol{y}}_t$ given all previous ones as well as the selected region is computed by

$$p(\boldsymbol{y}_t = \hat{\boldsymbol{y}}_t|\boldsymbol{y}_{<t}, R, \boldsymbol{a}_t) \equiv p(\boldsymbol{y}_t = \hat{\boldsymbol{y}}_t|\boldsymbol{h}_{t-1}, \boldsymbol{c}_t) = \text{softmax}(f_e(\boldsymbol{y}_{t-1}, \boldsymbol{h}_t; \boldsymbol{\theta}_e)), \tag{8}$$

where $f_e(\cdot)$ is a fully connected layer. An overview of the whole architecture is given in Fig. 2.

### 2.4 OPTIMISATION

The differentiation of $\ell$ w.r.t. all model parameters yields the following learning rule[3]

$$\frac{\partial \ell}{\partial \theta} = \sum_{t=1}^{N}\sum_{\boldsymbol{a}_t} p(\boldsymbol{a}_t|\boldsymbol{y}_{<t}, R)\left[\frac{\partial \log p(\boldsymbol{y}_t|\boldsymbol{y}_{<t}, R, \boldsymbol{a}_t)}{\partial \theta} + \log p(\boldsymbol{y}_t|\boldsymbol{y}_{<t}, R, \boldsymbol{a}_t)\frac{\partial \log p(\boldsymbol{a}_t|\boldsymbol{y}_{<t}, R)}{\partial \theta}\right],$$

where $\theta$ is a collection of model parameters, e.g., $\theta = \{\boldsymbol{\theta}_g, \boldsymbol{\theta}_s, \boldsymbol{\theta}_r, \boldsymbol{\theta}_i, \boldsymbol{\theta}_a, \boldsymbol{\theta}_e\}$ ($\boldsymbol{\theta}_s$ for the sharpener module). To reduce the computational cost as explained in Sect. 2.1, the derivative at time $t$ is often numerically approximated by the Monte Carlo method as follows

$$\frac{\partial \ell_t}{\partial \theta} \approx \frac{1}{S}\sum_{s=1}^{S} p(\hat{\boldsymbol{a}}_t^s|\boldsymbol{y}_{<t}, R)\left[\frac{\partial \log p(\boldsymbol{y}_t|\boldsymbol{y}_{<t}, R, \hat{\boldsymbol{a}}_t^s)}{\partial \theta} + \log p(\boldsymbol{y}_t|\boldsymbol{y}_{<t}, R, \hat{\boldsymbol{a}}_t^s)\frac{\partial \log p(\hat{\boldsymbol{a}}_t^s|\boldsymbol{y}_{<t}, R)}{\partial \theta}\right],$$

---

[2]Strictly speaking, the alignment in this modelling is no longer conditionally independent throughout the time as assumed in Sect. 2.1. In fact, this is the modelling used in both soft and hard attention mechanisms (Bahdanau et al., 2015; Xu et al., 2015). However, the discussion on why these mechanisms fail to achieve clear alignment still applies.

[3]We use the trick $\nabla_\theta p(x; \theta) = p(x; \theta)\nabla_\theta \log p(x; \theta)$ in the derivation.

where $S$ is the number of samples $\hat{\boldsymbol{a}}_t^s$ drawn from a categorical distribution defined by $p(\boldsymbol{a}_t|\boldsymbol{y}_{<t}, R)$ (Xu et al., 2015). Similar to Mnih et al. (2014) and Ba et al. (2015), this approximation yields a hybrid loss function asking for different optimisation strategies for the two terms in the square brackets. The former is optimised by a cross entropy loss together with the ground-truth token at time $t$ and gradient back-propagation. By regarding the accumulated sum of $\log p(\boldsymbol{y}_t|\boldsymbol{y}_{<t}, R, \hat{\boldsymbol{a}}_t^s)$ over the time as a reward, the latter is achieved by the REINFORCE algorithm (Williams, 1992). To reduce the high variance in gradient estimate caused by the unbounded $\log p(\boldsymbol{y}_t|\boldsymbol{y}_{<t}, R, \hat{\boldsymbol{a}}_t^s)$ (Ba et al., 2015), we follow Xu et al. (2015) to introduce a moving average baseline

$$b_j = 0.9 \times b_{j-1} + 0.1 \times \frac{1}{NS} \sum_{s=1}^{S} \sum_{t=1}^{N} \log p(\boldsymbol{y}_t|\boldsymbol{y}_{<t}, R, \hat{\boldsymbol{a}}_t^s),$$

where $j$ is the index of the mini-batch. Finally, we use the following learning rule for optimisation

$$\frac{\partial \ell}{\partial \theta} \approx \frac{1}{S} \sum_{s=1}^{S} \sum_{t=1}^{N} p(\hat{\boldsymbol{a}}_t^s|\boldsymbol{y}_{<t}, R) \left[ \frac{\partial \log p(\boldsymbol{y}_t|\boldsymbol{y}_{<t}, R, \hat{\boldsymbol{a}}_t^s)}{\partial \theta} + \lambda_r(\log p(\boldsymbol{y}_t|\boldsymbol{y}_{<t}, R, \hat{\boldsymbol{a}}_t^s) - b)\frac{\partial \log p(\hat{\boldsymbol{a}}_t^s|\boldsymbol{y}_{<t}, R)}{\partial \theta} \right],$$
(9)

where $\lambda_r$ is a learning hyper-parameter. We do not add entropy $H[\boldsymbol{a}_t]$ to (9) to further reduce gradient variance as in Xu et al. (2015), because it encourages a uniform distribution and breaks the alignment.

## 2.5 CONTEXT VECTOR

Rather than compute $\boldsymbol{c}$ from $X$ like both soft and hard attention do, the proposed sharp attention leverages the encoder output. Specifically, we explore three ways to compute $\boldsymbol{c}_t$ given $Z_t$, the encoder output at time $t$. The first one is *pooling*, where an average pooling of $Z_t$ is used for $\boldsymbol{c}_t$. Inspired by the glimpse idea (Mnih et al., 2014), we combine the result from pooling with $\hat{\boldsymbol{x}}_t^s$, the feature vector associated with the sampled region, to incorporate both fine and coarse representation of the target token, leading to the second way—*chain*. Alternatively, we may compute a weighted average of the feature set $\{Z_t, \hat{\boldsymbol{x}}_t^s\}$ for better mixed representation. With $\boldsymbol{h}_{t-1}$, the weights can be learned in a similar way to the soft attention as shown in Fig. 1(a). We call this last approach to $\boldsymbol{c}_t$ *weighting*.

## 3 RELATED WORK

In Xu et al. (2015), the alignment is treated as a latent variable to help define a loss function, which is optimised by gradually increasing a variational lower bound on marginal log-likelihood of the target output given the input sequence. Later, a variety of variational inference techniques (Lawson et al., 2018; Deng et al., 2018; Bahuleyan et al., 2018) are proposed to further reduce the gap between the lower bound and the marginal log-likelihood. Alternative approaches to approximating the marginal log-likelihood can be found in Shankar et al. (2018) and Shankar & Sarawagi (2019). The former cherry-picks a set of alignments, computes the log-likelihood conditioned on each alignment and averages the results, while the latter extends the idea by enforcing Markov property on adjacent alignments. Instead of approximation, Wu et al. (2018) attempted to compute exact marginal log-likelihood by assuming that each alignment is conditionally independent across time steps. All of the above methods regard the input sequence as fixed in optimisation and thus cannot tailor the input part(s) to clear alignment.

The idea of using relevant parts to improve attention has been explored in various tasks. Mei et al. (2016) adjusted the weights of the input parts resulting from soft attention to highlight the most relevant ones for selective generation. A similar work can be found in Nallapati et al. (2016), where keywords are interwoven with the sentences in which they lie for text summarisation by applying soft attention to sentences and words respectively and rescaling word weights. Instead of reweighting, Cheng et al. (2017) used character-level masks to guide the selection of useful parts for scene text recognition. None of these methods build clear alignment due to the use of soft attention.

Our work is closely related to Xu et al. (2015) and Ba et al. (2015). We generalise the former's mathematical formulation on hard attention by introducing flexible representation of objects of interest via a sharpener module. While the generalisation appears similar to the latter, we use it to tackle the alignment issue in attention mechanisms rather than develop a new Seq2Seq model. Our modelling also differs. Instead of seeking desired objects within the whole image, a divide-and-conquer scheme

is used to gradually narrow the search range for accurate localisation. Another difference in modelling is that in Ba et al. (2015) prediction will not happen until a series of localisation across predefined time steps whereas in our work that immediately follows localisation at each time.

# 4 EXPERIMENTS

We demonstrate the efficacy of the proposed method in two different scenarios of increasing difficulty: (i) synthetic handwritten digit recognition and (ii) real-world scene text recognition. Rather than strive for state-of-the-art results, the focus here is to highlight (i) the performance of Seq2Seq models can be boosted if attention mechanism really yields clear alignment, that is, paying attention to the target object, and (ii) the proposed sharp attention is an effective approach to reaching the goal. Therefore, our vanilla system is built upon off-the-shelf modules and was trained without sophisticated parameter tuning schemes. Below we describe some common choices for all scenarios.

**Implementation**    All images are converted to grey scale and resized to $100\times32$. A variant of VGG (Shi et al., 2017) is then used as the backbone to extract a $24\times1$ feature map from each resized image as well as create the set of regions, whose height is clipped to 32. The input sequence $X$ is obtained by splitting the feature map along its width, leading the dimensionality of each $x_i$ to be the feature map depth (i.e. 512). Before feeding it into the attention mechanism, we follow Shi et al. (2017) to further process $X$ to capture long-range contextual information with a 2-layer bidirectional LSTM, where each layer has a forward LSTM and a backward LSTM, each having 256 hidden units. The depth of the attention mechanism is set to 256 and so is the number of the hidden units of the associated LSTM, which runs over some time to predict the output sequence $Y$. The number of time steps is set to the maximum transcription length in each scenario. As in Sutskever et al. (2014), an end-of-sequence token is used to indicate the finish of prediction. The STN in the sharpener is composed of a localisation network, a grid generator and a sampler. Given a region, the localisation network, achieved by the one described in Liu et al. (2016), uses it to estimate an affine transformation, which is then used by the grid generator to place a set of control points on the region. By sampling the intensity value at each control point in a way similar to Shi et al. (2019), the sampler produces a patch of given size as the STN output. When multiple STNs are used, the output of the previous one is used as the input to the next one. The output of the last STN is plugged into the encoder in the sharpener to compute $Z$. The whole system was implemented using TensorFlow (Abadi et al., 2016) and the code will be released in the near future.

**Training**    Three kinds of Seq2Seq models were trained from scratch in terms of the attention mechanism used. Specifically, the soft and hard models were learned via corresponding attention mechanisms respectively. Unlike the previous two baseline models, the sharp models were obtained by the sharpener with the context vector schemes described in Sect. 2.5. A stochastic gradient descent method, ADADELTA (Zeiler, 2012), was used to learn the model parameters until certain number of iterations in different scenarios. The learning rate was constant and set to 1.0 and the decay rate was 0.95. In addition, all model parameters were regularised by an $L_2$ norm with a weight decay of $4\times10^{-5}$. All experiments were done with a batch size of 192 (per GPU) on a workstation of 4 NVIDIA GEFORCE RTX 2080 Ti GPUs. The number of samples $S$ was set to the batch size.

**Evaluation**    A prediction is correct if the predicted transcription matches ground truth. We reported the proportion of correct predictions on each testing dataset. As in Shi et al. (2017), all transcriptions were converted to lower cases and had punctuations ruled out before evaluation if applicable.

## 4.1 HANDWRITTEN DIGIT RECOGNITION

We randomly chose $l$ images from the MNIST dataset (Lecun et al., 1998), resized them to $32\times32$ and concatenated them horizontally, leading to an image of $l$ handwritten digits. For each $l$ in {5, 7, 9, 11, 13}, we created 20,000 images for training and 10,000 for testing by selecting from the MNIST training and testing datasets respectively, leading to a normal handwritten digit dataset. To introduce some distortion, we repeated the above procedure for a rotated dataset by randomly turning the selected images around $y$-axis within [-30°, 30°] before concatenation. Examples of the generated images are given in Fig. 3. We trained all models with the resulting datasets for 30,000 iterations by setting $\lambda_r = 1.0$ when applicable. For better localisation, we upsampled the set of regions along

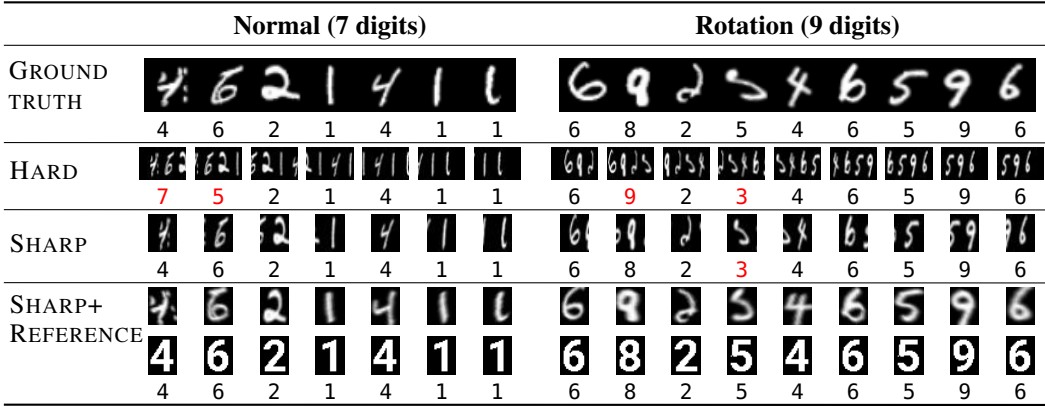

Figure 3: From top to bottom, we show examples of the created handwritten digit images and transcriptions (row 2), and patches for context vector computation as well as predicted digits (failures shown in red) at each time step (rows 3-5). Note that the patches are the receptive fields for the hard model and the output of the STN for the pooling based sharp models. The reference images are given below the patches in the last row.

Table 1: Recognition accuracy of all models for the handwritten digit datasets.

| Model | Normal | | | | | Rotation | | | | | Mean Acc. |
|---|---|---|---|---|---|---|---|---|---|---|---|
| | 5 | 7 | 9 | 11 | 13 | 5 | 7 | 9 | 11 | 13 | |
| **Baseline** | | | | | | | | | | | |
| SOFT | 97.2 | 96.6 | 95.0 | 91.9 | 82.2 | 96.3 | 95.4 | 93.6 | 89.1 | 78.3 | 91.6 |
| HARD | 97.2 | 96.4 | 94.1 | 91.6 | 81.7 | 96.6 | 95.4 | 93.5 | 89.2 | 78.6 | 91.4 |
| **Sharp** | | | | | | | | | | | |
| POOLING | 97.8 | 97.5 | 96.6 | 94.9 | 92.8 | 97.1 | 96.4 | 95.0 | 92.8 | 89.6 | 95.1 |
| CHAIN | 97.5 | 97.3 | 96.3 | 95.0 | 93.7 | 97.3 | 96.7 | 95.5 | 94.1 | 91.7 | 95.5 |
| WEIGHTING | 95.0 | 95.4 | 95.0 | 92.6 | 90.2 | 94.3 | 94.9 | 94.6 | 91.7 | 87.9 | 93.2 |
| **Pooling-based Sharp+Reference** | | | | | | | | | | | |
| AFFINE | **98.1** | **97.7** | **96.8** | **96.0** | **94.4** | **98.0** | **97.1** | **96.1** | **95.2** | **93.0** | **96.2** |

the width with a scale factor of 1.8 before plugging them into the sharpener, which was efficiently achieved by running region generation with 180×32 images. The patch output by the STN had a size of 24×32.

Table 1 shows that all sharp models significantly outperform the baseline, demonstrating the efficacy of clear alignment in attention mechanism. This can be easily seen from the pooling based model whose context vector purely results from the sharpener. Figure 3 also illustrates how attention can benefit from the sharpener. Take the rotation case for example. To predict digit '8' (the second column from left), hard attention chose a feature corresponding to a region filled with four digits. Due to the distraction of irrelevant digits (e.g., '6', '2' and '5'), the feature failed to precisely represent '8', thus giving wrong result. Although sharp attention selected the same region, it avoided most of the distraction by deliberately locating '8' in that region, thus leading to more accurate and specific representation as well as correct prediction. Besides, the resulting attention also has better interpretability since it is more focused. The above results testify that the performance of Seq2Seq models can be largely improved when attention mechanism is really focused on the object of interest.

To show that the sharpener allows for external supervision, a set of 24×32 reference images for each digit (see Fig. 3) was created with the Roboto Bold typeface of font size 36.[4] The supervision was achieved by introducing an $L_2$ image similarity loss of a weight of 1.0 to (9) to minimise the intensity difference between the patch and the reference. By showing what a desired digit would look like, the

---

[4]The font is available at https://fonts.google.com/specimen/Roboto. We used Pygame for rendering.

Table 2: Recognition accuracy of all models for the scene text recognition datasets.

| Model | IIIT | SVT | IC03 | IC13 | IC15 | SP | Mean Acc. |
|---|---|---|---|---|---|---|---|
| | 3000 | 647 | 867 | 857 | 1811 | 645 | |
| **Baseline** | | | | | | | |
| Soft | 77.9 | 78.8 | 87.8 | 86.1 | 61.4 | 62.5 | 75.8 |
| Hard | 77.6 | 78.8 | **89.2** | 86.0 | 59.9 | 63.9 | 75.9 |
| **Sharp+One STN** | | | | | | | |
| Chain | 76.9 | 78.1 | 88.7 | **88.8** | 61.1 | **65.6** | 76.5 |
| **Sharp+Two STNs** | | | | | | | |
| Pooling | 73.9 | 72.6 | 83.6 | 83.3 | 54.2 | 60.2 | 71.3 |
| Chain | 77.9 | **80.1** | 89.0 | 87.7 | **62.0** | 65.1 | **77.0** |
| Weighting | 77.7 | 78.5 | 89.0 | 87.4 | 61.5 | 64.0 | 76.4 |

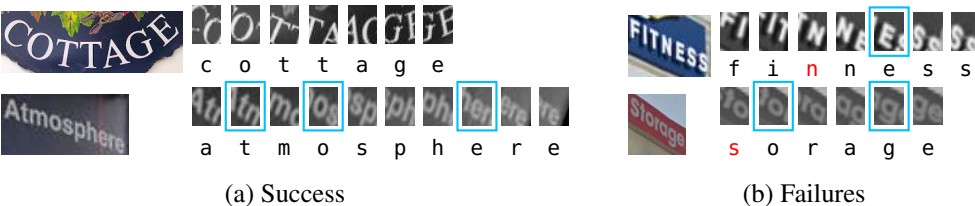

(a) Success                                    (b) Failures

Figure 4: Examples of recognition results for scene text recognition. All real-world testing images are shown as is without rescaling and grey scale conversion. We leverage the chain based sharp model to generate the patches localised by the sharpener and predicted tokens (failures highlighted in red) across time steps. The results from other sharp models look similar to what has been shown here.

pooling based sharpener gives better localisation where all digit shape is well preserved with little distraction, compared to its counterpart without such guidance (see Fig. 3). This, in turn, improves the representation of the context vector for alignment, leading to a boost of the model performance. Table 1 shows that such improvement is the most remarkable when handling images of more digits (e.g., 13). This is not surprising because the receptive fields in such case have more digits filled and thus make it difficult for the baseline models to decide where to look. Even if compared with the best one amongst all sharp models trained without external supervision, i.e. the chain based model, the improvement is still noticeable, further demonstrating the importance of accurate and target-specific representation in attention mechanism.

## 4.2 Scene Text Recognition

To further show the effectiveness of the proposed sharp attention, we applied it to a real-world visual sequence learning task, scene text recognition. The training datasets include MJSynth (Jaderberg et al., 2014) and SynthText (Jaderberg et al., 2016), while the testing ones consist of IIIT5K-Words (Mishra et al., 2012), Street View Text (Wang et al., 2011), ICDAR2003 (Lucas et al., 2003), ICDAR2013 (Karatzas et al., 2013), ICDAR2015 (Karatzas et al., 2015) and SVT Perspective (Phan et al., 2013), which are short for IIIT, SVT, IC03, IC13, IC15 and SP respectively. There are 8.9 million images in the MJSynth dataset and 5.5 million in SynthText by cropping text regions and ruling out non-alphanumeric characters. The number of images in each testing dataset is detailed in Table 2, where the three datasets, IC03, IC13 and IC15, were prepared by following the protocol in Baek et al. (2019). The total number of testing images is 7, 827. Note that some of these datasets (e.g., IC15 and SP) are quite challenging due to various nuisance factors, such as poor lighting and geometry change (Fig. 4).

For fast evaluation of various model configurations, we randomly sampled 2 million labelled images from the MJSynth dataset. The sampling was done by following the distribution (i.e. histogram of the transcription length) of the original dataset. We set the maximum transcription length to 16 to enable a large batch size (i.e. 192) for robust training. For sharp models, we upsampled the regions along the width with a scale factor of 2.0 for better localisation. To explore the effects of using multiple STNs for localisation, we first used two STNs to train all sharp models, and then just used

the second one to train a chain based model for comparison. To see whether localisation benefits
from a coarse-to-fine search strategy, the first STN was designed to estimate a simple transformation
(i.e. $x$-direction translation) but output a large patch (i.e. 64×32), while the second STN had the same
configuration as used in Sect. 4.1. All models were trained for 400,000 iterations with the sampled
dataset, and no reference images were used. To train the hard and sharp models we used $\lambda_r = 0.1$.

Table 3: Mean entropy of different attention mechanisms for the scene text recognition datasets.

| Model | IIIT | SVT | IC03 | IC13 | IC15 | SP | Mean |
|---|---|---|---|---|---|---|---|
| SOFT | 1.06 | 1.01 | 1.06 | 1.00 | 1.15 | 1.12 | 1.07 |
| HARD | 0.63 | 0.61 | 0.67 | 0.63 | 0.63 | 0.62 | 0.63 |
| SHARP | **0.42** | **0.36** | **0.42** | **0.39** | **0.38** | **0.38** | **0.39** |

Table 4: Sharp attention vs. soft attention in a published scene text recognition work.

| Model | IIIT | SVT | IC03 | IC13 | IC15 | SP | Mean Acc. |
|---|---|---|---|---|---|---|---|
| SOFT(Baek et al., 2019) | **84.3** | 83.8 | 93.1 | 91.9 | 70.8 | 71.9 | 82.6 |
| SHARP | 83.9 | **85.8** | **94.3** | **92.2** | **71.3** | **73.6** | **83.5** |

From Table 2, we see that the chain based one again works the best amongst all sharp models of two
STNs. It beats the baseline models remarkably on most of the datasets, whereas the other two either
moderately outperform or lag behind the baseline. It also beats its counterpart of single STN, even
though the latter is slightly better than the baseline as well. The result from the winning sharp model
further testifies our hypothesis on clear alignment in large-scale real-world datasets, which is also
revealed by Fig. 4. Whenever prediction is successfully performed, there is sensible localisation of
the target token. In fact, the sharpener attempts to highlight the token by placing it in the patch centre
(see Fig. 4 for examples surrounded by blue boxes). This is an encouraging result given that the
sharpener was trained in a data-driven manner with merely sequential labelling (i.e. transcriptions).
However, the localisation is by no means satisfactory since all patches have some sort of distractions,
such as skewed target tokens and irrelevant parts of adjacent tokens. Both this observation and the
benefit of multi-STN suggest a potential increase in accuracy if the sharpener is properly designed
such that it can produce good localisation as shown in Fig. 3, which is beyond the scope of this paper.

In Fig. 3, we have shown sharp attention is more focused and yields better interpretability. This can
be evaluated by $H[\boldsymbol{a}_t]$, entropy of the categorical distribution defined by $\boldsymbol{\alpha}_t$ (Shankar & Sarawagi,
2019). It measures attention uncertainty, that is, the lower entropy, the better alignment and thus
interpretability. Averaging $H[\boldsymbol{a}_t]$ across all valid time steps leads to the entropy for an image. We
reported the mean of such entropy on each testing dataset for various attention mechanisms in
Table 3. The results clearly show that sharp attention indeed boosts interpretability since entropy is a
logarithmic metric. We only reported the entropy from the best sharp model in Table 2.

Finally, we used the full datasets (i.e. MJSynth & SynthText) to train a sharp model with the same
configuration to the best one in Table 2. To fairly compare the proposed attention with other attention
mechanisms in existing scene text recognition works, we reported the performance of the sharp model
and a model (i.e. VGG+BiLSTM+Attn) based on soft attention trained with the same datasets by
Baek et al. (2019) in Table 4. The two models share the same backbone and RNN decoder. They only
differ in attention mechanism. Table 4 further shows the superiority of our method.

## 5  CONCLUSION

We have described a novel attention mechanism that is able to build clear alignment between relevant
regions in the input image and the target output. This is achieved by a generic sharpener module that
computes accurate representation of the targets across time steps. Experimental results show that a
vanilla implementation of our method can significantly beat soft and hard attention on both synthetic
and real-world datasets in terms of performance and interpretability, without bells and whistles such
as the auxiliary model in Ba et al. (2015) and sophisticated training schemes in Xu et al. (2015). We
plan to apply our method to more visual sequence learning tasks in the future.

## 6 REPRODUCIBILITY STATEMENT

The implementation details have been given in lines 185–202. Although most of the training parameters have been described in lines 207–212, some task-specific setting can be found in lines 223–227 and 266–274 respectively. The generation of synthetic handwritten digit dataset has been detailed in lines 217–222, while the preparation of real-world scene text recognition datasets has been elaborated in lines 254–261.

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
