# OpenReview forum: "Sharp Attention for Sequence to Sequence Learning"
_ICLR.cc/2022/Conference — ICLR 2022 Submitted_

### Official Review · Reviewer_5Hay · 2021-10-25

**Correctness:** 3
**Technical Novelty And Significance:** 2
**Empirical Novelty And Significance:** 2
**Recommendation:** 6
**Confidence:** 4

**Main Review:**

Strengths:

The alignment is one issue in the normal attention mechanism, especially for the long sequence. In this paper, the authors propose a sharpener module, which is explicitly designed to seek clear alignment. The motivation is strong, and the idea performs well on the two handwritten recognition tasks.

Weakness

(1) The paper organization maybe is not friendly to read. For example, as the sharpener module is the main part of the paper, the description on this part is not concentrated, Some are in the Section 2.3, and some are in the section 2.5, and some are in the section 4 (Implementation). I suggest the authors can merge all details to create one new section to describe sharpener module. If the paper size is limited, maybe the section 2.4 can be shorten because I think this part is very similar with the previous work.

(2) Some details are not clear. For example, it is not clear to know how the three ways to compute context vector c given Z. Also how to get the r_i in the Fig2 (choosing the region regarding to the largest value alpha_{t,i}?).

(3)The experiments maybe are not sufficient. I am not sure what kind of SOFT attention is used in he baseline. Maybe some kind of improvement on SOFT attention can be compared in the experiments, for example some works introduced in Related work.

(4)The experiments are not clear. Because some datasets in the scene text recognition are curved images, some works integrated the STN module in the recognition. So if the baseline system is not integrated with STN, while the proposed module uses STN, then this comparison is not fair. Maybe you should use the baseline system integrated with STN.

(5) If the authors can do the experiments on the real handwriting recognition, neither English or Chinese text, this will be much better, because the digits are not connected in synthetic handwritten digit string , and the sequence length is not very long in the scene text.

**Summary Of The Paper:**

In this paper, the authors propose a sharpener module to seek clear alignment in the attention mechanism. The sharpener module contains a localiser and an encoder, where the localiser is used to adjust the interested region to align the target better, and the encoder is used to extracted features from the adjusted region, which is used to get the final context vector.

The authors verify the proposed method on two datasets, including synthetic handwritten digit recognition, and one real-world scene text recognition.  The experimental results show the effectiveness of the proposed method. In addition, the authors show the detailed analysis on the results, which verify the proposed idea further.

In summary, the idea is well motivated, and the experimental results show its effectiveness.

**Summary Of The Review:**

This paper is well motivated to seek clear alignment in the attention mechanism. This idea can be understood overall, although some details can be added. For the experiments, it seems not sufficient, maybe more comparison experiments can be added to verify the effectiveness of the proposed method.

---

### Official Review · Reviewer_1RPC · 2021-10-31

**Correctness:** 3
**Technical Novelty And Significance:** 2
**Empirical Novelty And Significance:** 2
**Recommendation:** 3
**Confidence:** 4

**Main Review:**

Strengths of this paper:
(1). A new sharp attention mechanism is proposed, by introducing a latent vector into traditional attention framework, so that let the target output token only depends on one element of input sequence.
(2). The general mathematical design and analysis of the sharp attention idea seems reasonable and feasible.

Weaknesses:
(1) The proposed method can be somehow regarded as a variant of hard attention, since the authors assume that the output y_t only depends on a_t at time t (line 55-56). I don't think to pursue the alignment such that each target token only depends on one element of the input X is a better way comparing with traditional soft attention mechanism. The reason is two-fold. At one hand, such sharp attention may require that each of local region (broken down by some method, such as sliding window) should large enough to cover objects of interest, however, in practice, we can not always guarantee this since sometime we need to use small region in order to make better attention on small instance in a sequence. On the other hand, contextual information is crucial for some sequence recognition problem like handwritten or scene text recognition. Sharp attention may lose the ability to make use of contextual information which is very helpful during the recognition process.

(2) The implementation of the Sharpener in this paper using STN plus CNN. Actually, STN was designed to crop and transform (eg. under affine transformation) an input image so that the output of STN becomes more regular, although its localization subnet can learn a set of fiducial points that seem localize the real instance of interest. In the field of scene text recognition, STN  has been successfully used as a rectification by many previous methods, together with an attention Seq2Seq recognizer, such as "Shi, et al., Robust Scene Text Recognition with Automatic Rectification, CVPR 2016". Therefore, it is doubtful that the performance gain achieved by the proposed Sharpener actually comes from the STN rectification rather than the so called sharpen attention.

(3) The experiments are not solid and convincing enough:
   - The first experiment using synthetic handwritten digital string is quite simple and toy. To show the proposed method can be really useful, I recommend the authors should use more challenging dataset, such as IAM dataset which is widely used in the community of OCR.
   - I don’t understand why in section 4.2, the authors just used a sampled 2 million samples from MJSynth dataset to train the different models for comparison. Experiments in this paper are not heavy so I see no reason to use just part of the dataset to train the models. To use either of both of MJSynth and SynthText datasets like existing methods will give more accurate evidence to show if the proposed SHARP model works better than the counterpart. Otherwise, I am wondering if the proposed method only performs better under small training sample setting.
   - More strict comparison with related methods are expected under the same settings, such as:
        * B. Shi, et al., Robust Scene Text Recognition with Automatic Rectification, CVPR 2016;
        * T. Wang, et al., Decoupled attention network for text recognition, AAAI 2020；
        * Q. Wang, et al., ReELFA: A scene text recognizer with encoded location and focused attention, ICDAR 2019

(4) It is unclear that the proposed Sharpener module is a general module that can be applied in different existing SOTA Seq2Seq methods, such as ATSTER in the field of scene text recognition, and so on.

(5) It is better to illustrate the attention map generated by the proposed method when the recognition results are not correct, so that we can see that if the proposed method can solve the problem of attention mis-alignment issue in such circumstance.


**Summary Of The Paper:**

This paper presents a new module technology for seq2seq attention model, addressing the mis-alignment issue which widely exist in most current attention-based methods. A new Sharpener module is proposed to help construct clear (direct) alignment in attention mechanism. Mathematical design and analysis of this idea is given in detail, and the authors also give an implementation of Sharpener module, which consist of a localizer (STN) and a n encoder (CNN), under the design concept that the localizer locates the target in the region and the encoder extracts better feature for alignment. Equipped with the proposed Sharpener module, a new Seq2Seq model is proposed for text recognition. Experiments on both synthetic handwritten digital string and real-world scene text datasets show the superiority of the proposed method, comparing with its counterpart, i.e., traditional hard or soft attention-based methods.

**Summary Of The Review:**

In general, this problem addressed in this paper is not a new one. There are many existing methods have been proposed to address the attention mis-alignment (or attention drift) issue. Although the motivation of this paper and the mathematical design of the sharp attention mechanism seem ok and reasonable, the proposed new Sharpener module does not bring me too much new insight and inspiration.  I do not see clear advantage of the proposed method against existing ones. Furthermore, I am not satisfied with the experimental part of this paper.

---

### Official Review · Reviewer_aDJW · 2021-11-02

**Correctness:** 3
**Technical Novelty And Significance:** 3
**Empirical Novelty And Significance:** 3
**Recommendation:** 5
**Confidence:** 3

**Details Of Ethics Concerns:**

N.A.

**Main Review:**

This paper is well-written and easy to follow. The motivation is clear. The key innovation is the sharpener module for better feature alignment, which theoretically sounds according to the equations in the manuscript. Experimental results show that the proposed method consistently outperforms the original attention mechanism over different datasets on scene text recognition task.

My major concerns are as follows:
1)	This method is claimed to be a novel attention mechanism for sequence-to-sequence learning. As such, more experiments on other tasks are expected (except for the scene text recognition task) for a better demonstration of the proposed method. Otherwise, I’m thinking it’s better to claim that this method is designed for scene text recognition task.
2)	For experiments in Fig. 3 and Table 1, what’s the motivation of conducting experiments with reference patches? Why not conduct such experiments for experiments in Tables 2 and 4?
3)	As the experiments are mainly for scene text recognition task, I’m wondering if the proposed method could help on scene text recognition task. Currently, the performances shown in Table 4 are very far from state-of-the-art.


**Summary Of The Paper:**

This paper presents a sharp attention module for sequence-to-sequence learning. It introduces a sharpener to compute context vectors for clearer alignment of features. Extensive experiments on scene text recognition task demonstrate the effectiveness of the proposed method.

**Summary Of The Review:**

In summary, this paper is well-written and its key contribution technically sounds. Extensive results on the scene text recognition task show the effectiveness of the proposed method as compared with the traditional attention mechanism. However, the experiments are not thorough and enough to support the authors’ claim well. Additionally, some details of the experiments are not very clear to me.

---

### Official Review · Reviewer_mESb · 2021-11-03

**Correctness:** 2
**Technical Novelty And Significance:** 2
**Empirical Novelty And Significance:** 2
**Recommendation:** 3
**Confidence:** 4

**Main Review:**

Strengths:
The attention machanism with a sharpener module is better than the baseline attention mechanisms.

Weaknesses:
The experimental results are not convincing enough. The results on the scene text recognition benchmark are much lower than the current mainstream results [1,2,3]. There are more than 10 percent gaps on these datasets. Thus, the experimental results are not sufficient to prove the effectiveness of the sharpener module since the baseline results are not convincing.


[1] Shi, Baoguang, et al. "Aster: An attentional scene text recognizer with flexible rectification." IEEE transactions on pattern analysis and machine intelligence 41.9 (2018): 2035-2048.
[2] Qiao, Zhi, et al. "Seed: Semantics enhanced encoder-decoder framework for scene text recognition." Proceedings of the IEEE/CVF Conference on Computer Vision and Pattern Recognition. 2020.
[3] Yan, Ruijie, et al. "Primitive Representation Learning for Scene Text Recognition." Proceedings of the IEEE/CVF Conference on Computer Vision and Pattern Recognition. 2021.

**Summary Of The Paper:**

This paper proposed a sharpener module for sequence-to-sequence learning. The motivation is to address the construction of clear alignment in attention mechanism. The authors conducted experiments on handwritting text recognition and scene text recognition text datasets. The experimental results show that the attention mechanism with a sharpener module is better than the soft attention and hard attention.

**Summary Of The Review:**

The major concern of this paper is that the baseline results are too bad, making the experimental result not convincing. I suggest the authors choose a SOTA method as baselines.

---

### Decision · Program_Chairs · 2022-01-20

**Decision:**

Reject

**Comment:**

The paper proposed a sharp attention mechanism in the context of image to sequence modeling. It seeks to build a “clear” alignment from the attention in order to improve the performance of the task. I don’t think there is a general consensus in the research community that the “clear” or “hard” attention performs better than the vanilla “soft” attention. Therefore, experiments become the key in justifying such motivation (and the model). However, as all the reviewers point out, the experiments in this paper are not satisfying. The numbers are far from the current mainstream results (Reviewer mESb). The experiments are done on relatively small datasets/tasks (Reviewer 1PRC) and the comparisons aren’t strictly speaking fair (Reviewer 5Hay). I think this alone is enough reason for the rejection of this paper.

Additionally, on the algorithmic side, the novelty of this mechanism is not that high, given the existence of work such as the hard attention (Xu et al, 2015) and variational attention (Deng et al, 2018). It is also an unanswered question how such a mechanism can be introduced into the modern architectures that use attention (e.g. the multi-head attention). The authors did not respond to the questions of the reviewers.